# Characterization and Anti-Inflammatory Effects on Periodontal Ligament Cells of *Citrus limon*-Derived Exosome-like Nanovesicles Under Different Storage Temperatures

**DOI:** 10.3390/biomedicines14010099

**Published:** 2026-01-03

**Authors:** Yiming Ma, Chenhao Yu, Guojing Liu, Jia Liu, Qingxian Luan

**Affiliations:** Department of Periodontology, Peking University School and Hospital of Stomatology, National Center of Stomatology and National Clinical Research Center for Oral Diseases, National Engineering Laboratory for Digital and Material Technology of Stomatology, Beijing Key Laboratory of Digital Stomatology, Beijing 100081, China; 1810303111@pku.edu.cn (Y.M.);

**Keywords:** *Citrus limon* L., plant-derived exosome-like nanoparticles, storage temperature, stability, periodontal ligament cells, inflammatory cytokines

## Abstract

**Objectives**: The purpose of this study is to compare the differences between lemon-derived exosome-like nanovesicles (LELNs) stored at −80 °C, −20 °C, and 4 °C for one month and freshly isolated LELNs, in terms of characterization and anti-inflammatory effects on periodontal ligament cells, aiming to identify suitable storage conditions for LELNs. **Methods:** Nanoparticle tracking analysis (NTA), transmission electron microscopy (TEM), and micro bicinchoninic acid assay (BCA) were conducted to access the characterization differences. LPS-induced human periodontal ligament cells were used as an in vitro inflammatory model, and the changes in biological functions were examined by qRT-PCR and ELISA. **Results:** LELNs stored at −80 °C retained the highest particle and protein concentration and showed the least aggregation and heterogeneity in size on TEM images, while the average particle sizes shown by NTA were similar. And LELNs exhibited similar anti-inflammatory effects on periodontal ligament cells after one month of storage at −80 °C, −20 °C, and 4 °C. **Conclusions:** We found that LELNs can maintain in vitro anti-inflammatory ability when stored at either −80 °C, −20 °C, or 4 °C for one month, while storing at −80 °C maintains the concentration and uniform particle size best.

## 1. Introduction

Extracellular vesicles (EVs) are universal terms for a diverse population of nanoscale membrane vesicles actively released by cells. They can be further categorized into three subgroups based on their biogenesis mechanisms: exosomes, microvesicles, and apoptotic bodies [1]. Apoptotic bodies are large vesicles formed during cell death with a diameter greater than 1 μm; cells release these vesicles when they undergo apoptosis [2]. Microvesicles are larger vesicles ranging from 100 to 1000 nm in diameter, generated through outward budding of the plasma membrane [3]. In contrast, exosomes are smaller vesicles, typically 30–150 nm in diameter. Unlike microvesicles and apoptotic bodies, exosomes are formed via inward budding of the plasma membrane, involving pathways such as the endolysosomal pathway, intraluminal budding of multivesicular bodies (MVBs), and the fusion of MVBs with the plasma membrane [4]. Exosomes carry cargos such as RNAs, proteins, and lipids, enabling them to mediate intercellular communication and regulate the biological behavior of target cells [5].

Plant-derived exosome-like nanoparticles (PELNs) are nanovesicles isolated from plants that share many similarities with mammalian exosomes, such as size distribution, morphology, surface charge, density, and certain components [6]. PELNs also contain various components that regulate physiological processes such as proteins, lipids, miRNA, etc. [7]. PELNs can be internalized by mammalian cells through endocytosis and fusion, thereby releasing the cargo carried within the PELNs [8]. Numerous studies have shown that PELNs possess various physiological functions, including anti-inflammatory [9,10], modulating microbiota [11], and regeneration [12,13]. Lemon-derived exosome-like nanoparticles (LELNs) are nanovesicles isolated from *Citrus limon* L. Many studies have demonstrated that LELNs exhibit multiple biological activities, such as anti-inflammatory [14], antioxidant [15,16], inhibition of cancer cell proliferation [17,18], and regulation of gut microbiota [19,20]. In addition, LELNs can be used as a drug delivery system; LELNs loading with doxorubicin can efficiently overcome cancer multidrug resistance [21]. Therefore, LELNs hold broad application prospects. Periodontitis is a chronic infectious disease caused by dental plaque microbes, leading to the inflammation and destruction of periodontal supporting tissues. Given their anti-inflammatory, antioxidant, and regenerative capacities, PELNs represent a potential therapeutic approach for periodontitis. The effect of LELNs on periodontal tissues remains understudied, and research is needed to investigate whether LELNs can play a role in treating periodontitis and to determine optimal storage conditions for LELNs that maintain their bioactivity and function after isolation.

In practice, a batch of isolated EVs cannot typically be used at once for experiments or clinical applications. Scaling up EVs for broader clinical applications poses a greater challenge for their storage. Therefore, identifying optimal storage conditions warrants significant attention. There is much evidence to suggest that different storage conditions have a great influence on EVs’ concentration, characterization, and biological functionality [22,23,24]. Temperature is the factor with the most influence. For long-term preservation (more than one week), EVs stored at −80 °C maintain their concentration, morphology, and biological function best [25,26,27]. And for short-term preservation (less than one week), storing at 4 °C is also a suitable choice to retain the characterization and function of EVs [28,29]. The freeze–thaw cycle is also an important factor; multiple freeze–thaw cycles lead to structural disruption, decreased concentration, and loss of content [30,31,32]. In addition, the use of cryoprotectants, lyophilization, storage containers, and pH can also affect the storage outcome of EVs [33,34,35,36,37].

Compared to mammalian-derived EVs, PELNs often contain plant sterols and flavonoids, which may exhibit unique stability characteristics under different storage conditions [38,39,40]. But studies about the storage of PELNs are limited and exhibit strong heterogeneity. *Kaempferia parviflora* extracellular vesicles can be stably maintained at −20 °C and −80 °C for 8 weeks, and lipid degradation occurs quickly at 4 °C and room temperature [41]. Meanwhile, leaf-derived extracellular vehicles combined with TMO and ginger-derived lipid vehicles remain stable when stored at 4 °C for about one month [42,43]. Thus, further studies on the optimal storage conditions of PELNs are needed.

We hypothesized that suboptimal storage conditions may induce physical instability in PELNs, which may directly compromise their biological functions. So in this study, we use lemon-derived exosome-like nanoparticles (LELNs) as an example to investigate changes in particle size, concentration, morphology, and anti-inflammatory effects on periodontal ligament cells after one month of storage under different temperature conditions, aiming to identify the most suitable storage conditions for LELNs.

## 2. Materials and Methods

### 2.1. Isolation of LELNs

Fresh lemons (*Citrus limon* L., sourced from Anyue County, Ziyang, China) were washed in PBS (pH = 7.4) (EallBio, Beijing, China). The lemon juice was extracted by mechanical squeezing. The juice was then centrifuged (Beckman Coulter, Brea, CA, USA) at 4 °C and 10,000× *g* for 30 min. The supernatant was collected, and the precipitate was discarded. This step was repeated three to four times until no visible precipitate remained. The final supernatant was then ultracentrifuged (Beckman Coulter, Brea, CA, USA) at 4 °C and 100,000× *g* for 70 min. After ultracentrifugation, the supernatant was discarded, and the pellet was retained. The pellet was resuspended in PBS, and the resuspension was filtered through a 0.22 μm sterile filter (Merck Millipore, Burlington, MA, USA). The filtered solution was collected and divided in 1.5 mL polypropylene centrifuge tubes and stored at −80 °C, −20 °C, and 4 °C for subsequent experiments.

### 2.2. Cell Culture

Human periodontal ligament cells (hPDLCs) were obtained from EallBio (Beijing, China). The tissues were cultured in α-minimal essential medium (α-MEM) (Gibco, Waltham, MA, USA) supplemented with 10% fetal bovine serum (Procell, Hong Kong, China) and 1% penicillin-streptomycin (EallBio, Beijing, China) in a 37 °C incubator with 5% CO_2_. hPDLCs from passages three to eight were used for subsequent experiments. The cell culture medium was changed every two to three days.

In the in vitro anti-inflammatory cell experiments, cells were divided into the following groups: control group, LELNs group, LPS group, and LPS + LELNs group. Lipopolysaccharide (LPS) (Invivogen, San Diego, CA, USA) was added into the medium at a concentration of 1 μg/mL for 24 h in the LPS group. LELNs were added into the medium at a concentration of 1 × 10^9^ particles/mL for 24 h in the LELNs group. And in the LPS + LELNs group, LPS was added into the medium at a concentration of 1 μg/mL for 2 h, followed by stimulation with 1 × 10^9^ particles/mL LELNs for 22 h. The determination of LELN concentrations and the dose–response curve (Appendix A) are presented in the Appendix A. Cells and cell culture supernatants from all groups were collected for subsequent experiments.

### 2.3. LELNs Uptake Assay

For the uptake assay, 90 μL of diluent C (provided with PKH26 dye, Beijing Fluorescence Biotechnology, Beijing, China) was mixed with 10 μL of PKH26 dye (Beijing Fluorescence Biotechnology, Beijing, China) to prepare a working solution. LELNs were incubated in this solution for 30 min, and then ultrafiltrated to remove unbound dye. PKH26-labeled LELNs were co-cultured with hPDLCs for 12 and 24 h. Subsequently, the cells were stained with Hoechst 33342 (Solarbio, Beijing, China) to label the cell nucleus. Then, the cells were fixed with 4% paraformaldehyde (PFA), permeabilized by 0.1% Triton X-100 (Solarbio, Beijing, China), and stained with Actin-Tracker Green-488 (Beyotime, Haimen, China) to label the cytoskeleton. Observations were made using confocal fluorescence microscopy (Leica, Wetzlar, Germany).

### 2.4. Transmission Electron Microscopy (TEM)

The morphology of LELNs was observed using transmission electron microscopy (TEM) (JEM-1400PLUS, JEOL, Tokyo, Japan). Approximately 10 μL of LELNs were added onto a copper grid and allowed to settle for 2 min. The sample was then negatively stained with 2% (*w*/*v*) phosphotungstic acid for 3 min, and then air-dried at room temperature for 10 min. The sample was observed under the TEM to capture images.

### 2.5. Nanoparticle Tracking Analysis (NTA)

The size distribution and concentration of LELNs were analyzed using nanoparticle tracking analysis (NTA) (NS300, Malvern Panalytical, Malvern, UK).

### 2.6. Micro Bicinchoninic Acid Assay (BCA) Analysis

The total protein concentrations of LELNs were measured by micro bicinchoninic acid assay (BCA) according to manufacturer’s instructions (Beyotime, Haimen, China).

Absorbance at 562 nm was determined using a SpectraMax ABS Plus Absorbance Microplate Reader (Molecular devices, Shanghai, China).

### 2.7. Quantitative Real-Time Polymerase Chain Reaction (qRT-PCR)

Total cellular RNA was extracted using the *SteadyPure* RNA Extraction Kit (ACCURATE BIOTECHNOLOGY (HUNAN) Co., Ltd., Changsha, China). The concentration and purity of the RNA were determined using a NanoDrop8000 (Thermo Fisher Scientific, Waltham, MA, USA). Subsequently, the mRNA was reverse transcribed into cDNA using the *Evo M-MLV* RT Premix for qPCR (ACCURATE BIOTECHNOLOGY (HUNAN) Co., Ltd., Changsha, China). Then, quantitative real-time polymerase chain reaction (qRT-PCR) (QuantStudio 1, Thermo Fisher Scientific, Waltham, MA, USA) was performed with SYBR Green Premix *Pro Taq* HS qPCR Kit (Rox Plus) (ACCURATE BIOTECHNOLOGY (HUNAN) Co., Ltd., Changsha, China) to analyze the expression of IL-6, IL-1β and TNF-α, normalized to GAPDH. The sequences of the primers are listed in Table 1 (Sangon, Shanghai, China). The relative gene expression level was analyzed by the comparative 2^−ΔΔCt^ method.

### 2.8. Enzyme Linked Immunosorbent Assay (ELISA)

Cell culture supernatants were collected and centrifuged at 4 °C and 1000× *g* for 10 min. The enzyme linked immunosorbent assay (ELISA) kits were obtained from Solarbio (Beijing, China), and were used according to the manufacturer’s instructions. Absorbance at 450 nm was determined using a SpectraMax ABS Plus Absorbance Microplate Reader (Molecular devices, Shanghai, China).

### 2.9. Cell Viability Assay

The CCK-8 kit (Beyotime, Haimen, China) was used to access cell viability and the cytotoxic effects of LPS. Cells were cultured in a 96-well template and treated with different concentrations of LPS for 24 and 48 h. Subsequently, 100 µL of working solution was added to each well, followed by incubation at 37 °C for 1 h. Absorbance at 450 nm was determined using a SpectraMax ABS Plus Absorbance Microplate Reader (Molecular devices, Shanghai, China).

### 2.10. Statistical Analysis

Statistical analyses were conducted with the use of GraphPad Prism v10.2.3 (GraphPad, San Diego, CA, USA). The statistical results in this study were expressed as means ± SD. One way ANOVA and Tukey’s multiple comparisons tests were used to compare the differences between groups. For comparisons across multiple groups, Dunnett’s multiple comparisons test was applied for post hoc pairwise analysis against the control group. A *p* value < 0.05 was considered statistically significant.

## 3. Results

### 3.1. Isolation and Characterization of LELNs

We used a simplified differential ultracentrifugation method developed by our research group to facilitate simple and efficient extraction of PELNs [44]. Initially, fresh lemons were squeezed to obtain the juice. Then, multiple rounds of high-speed centrifugation were applied to remove large particles and ultracentrifugation was applied to eliminate small particles. The pellet obtained from ultracentrifugation was resuspended in an appropriate amount of PBS. Finally, a 0.22 µm sterile filter was used to remove larger impurities and to obtain sterile LELNs for subsequent experiments (Figure 1). Using this method, we were able to extract about 3 × 10^12^ particles LELNs from 1 kg fresh lemons.

The morphology of LELNs was observed using TEM. As shown in Figure 2a, the LELNs exhibited a typical cup-shaped or spherical morphology with a size of approximately 100 nm. NTA results (Figure 2b) indicated that the LELNs had an average particle size of 142.5 ± 2.4 nm and a concentration of approximately 2.8 × 10^12^ particles/mL. PKH 26-labeled LELNs were co-cultured with hPDLCs for 12 and 24 h, and we can see from Figure 2c that LELNs can be internalized by hPDLCs after 12 and 24 h.

### 3.2. Effects of Different Storage Temperatures on the Characterization of LELNs

LELNs were stored at −80 °C, −20 °C, and 4 °C for one month, and the following experiments were conducted.

As shown in Figure 3a and Table 2, the results of NTA revealed that there was no significant difference in average particle size and the size distribution of LELNs among all groups. NTA also showed that the concentration of freshly isolated LELNs was 2.8 × 10^12^ particles/mL, and the concentrations of LELNs stored at −80 °C, −20 °C, and 4 °C were, respectively, 2.6 × 10^12^, 2.0 × 10^12^, and 1.8 × 10^12^ particles/mL (Table 2). This indicates that LELNs decreased across all storage temperatures compared to freshly isolated LELNs, with the LELNs stored at 4 °C showing the lowest concentration.

The total protein concentration of LELNs was measured using a BCA protein assay kit. As shown in Figure 3b, the total protein concentration of freshly isolated LELNs was 6.84 mg/mL. The total protein concentration of LELNs stored at −80 °C for one month was 6.32 mg/mL, showing no significant difference from the freshly isolated LELNs. In contrast, the total protein concentrations of LELNs stored at −20 °C and 4 °C for one month were 5.82 mg/mL and 5.32 mg/mL, respectively, both showing a significant decrease compared to the freshly isolated LELNs.

The morphological changes in LELNs under different storage temperatures were observed using TEM. As shown in Figure 3c, cup-shaped or spherical lipid bilayer nanovesicles were observed in all groups. The freshly isolated LELNs and those stored at −80 °C for one month exhibited relatively uniform sizes, around 100 nm. In contrast, both the LELNs stored at −20 °C and those stored at 4 °C for one month showed aggregation and heterogeneity in particle sizes, with some particles significantly enlarged, reaching approximately 200 nm.

### 3.3. LELNs Stored at Different Temperatures Retmain In Vitro Anti-Inflammatory Effects

LELNs were stored at −80 °C, −20 °C, and 4 °C for one month and diluted to the same concentration according to the NTA for the following experiments.

The relative mRNA levels of IL-6, IL-1β, and TNF-α in hPDLCs under different treatments were detected using qRT-PCR. As shown in Figure 4a–c, LELNs stored at −80 °C, −20 °C, and 4 °C for one month did not exhibit any pro-inflammatory effects. Instead, they all significantly reduced the mRNA levels of LPS-induced pro-inflammatory cytokines. Among them, the LELNs stored at −80 °C demonstrated the best anti-inflammatory effect.

The protein expression levels of the pro-inflammatory cytokines IL-6, IL-1β, and TNF-α in the supernatants of hPDLCs under different treatments were measured using ELISA. As shown in Figure 4d–f, LELNs stored at −80 °C, −20 °C, and 4 °C did not increase the levels of inflammatory cytokines (IL-6, IL-1β, TNF-α) in the supernatant of hPDLCs, but significantly reduced their secretion induced by LPS.

## 4. Discussion

Comparing to mammalians-derived EVs, PELNs have advantages with regard to large-scale production, low toxicity, low immunogenicity and high biocompatibility, making PELNs a potential substitute as therapeutic drug delivery vehicles [39,40]. However, large-scale production and application inevitably raise the challenge of better storage to maintain morphology and physiological functions similar to freshly isolated PELNs. Nevertheless, existing research on the storage of PELNs is still very limited. Therefore, this study aims to use LELNs as an example to investigate the optimal storage conditions for PELNs.

In this study, we found that compared to storage at −20 °C and 4 °C, LELNs LELNs stored −80 °C maintained the highest particle concentration and protein concentration, supported by several previous studies [45,46,47]. NTA showed that the average LELNs sizes remained similar after storage at −80 °C, −20 °C, or 4 °C. Meanwhile, TEM images showed that storage at −20 °C and 4 °C can lead to increased and varied particle sizes, and aggregation was observed in all three groups. Given the technical limitations of both NTA and TEM, the morphological features of LELNs appear largely comparable across the different storage temperature conditions. It is commonly believed that ice crystals are formed when EVs are rapidly cooled, and the formation of ice crystals damages the phospholipid bilayer, resulting in changes to the physical characteristics of EVs [22,40]. However, several previous studies have reported results consistent with our findings. Cardiac progenitor cells-derived EVs retain similar average sizes after being stored for one week at 4 °C and −80 °C [28]. MSC-TNFR EVs stored for 20 weeks did not reveal any consistent storage buffer-related or temperature-related differences in shape, diameter, or intactness on TEM images [48]. These findings suggest that even in the presence of ice crystals, the structure of EVs may not undergo gross alterations during short- or long-term storage, which may be related to the type of EVs.

Furthermore, we observed that the anti-inflammatory effects on periodontal ligament cells of LELNs were retained regardless of whether they were stored at −80 °C, −20 °C, or 4 °C. Several previous studies have also reported similar phenomena. Cardiac progenitor cells-derived EVs retained in vitro migration functionality when stored at either 4 °C or −80 °C for 1 week [28]. Exosomes from human whole saliva can retain membrane stability after storage at 4 °C for 20 months [49]. EVs from Kaposi’s sarcoma-associated herpesvirus (KSHV)-infected human endothelial cells showed similar stability and biological activity at 4 °C and −80 °C until day 25 [47]. However, there is a lack of in vitro functional evaluation of stored LELNs, and few studies have investigated the effects of stored EVs on periodontal cells. Our study suggests that although concentration decreased in −20 °C and 4 °C groups, the active components responsible for the anti-inflammatory effects on periodontal ligament cells may have been retained, potentially due to a reduction in structural proteins or other proteins unrelated to anti-inflammatory capacity. This also indicates that PELNs could potentially serve as a stable medium for storing bioactive substances. Future studies may involve extending the storage period to observe the long-term functional stability of LELNs.

As for the limitations of this study, we focused solely on the effects of different storage temperatures on LELNs, without considering factors such as freeze–thaw cycles, pH, storage medium, or storage containers. However, we controlled variables to the greatest extent possible. All LELNs samples were aliquoted and stored in 1.5 mL polypropylene centrifuge tubes after the isolation process, and underwent only a single freeze–thaw cycle prior to characterization tests and in vitro experiments. Additionally, PBS was used as the solvent throughout, with no cryoprotectants added. However, it must be acknowledged that the storage duration in this study was relatively short, for only one month. Although we were able to identify early degradation trends, determining the long-term stability and functional shelf-life of LELNs is essential for clinical translation and industrial application, which will be the focus of our subsequent work. Future studies may also involve the application of cryoprotectants or lyophilization for LELNs storage. In addition, this study only focused on the changes in the anti-inflammatory activity of stored LELNs in vitro. Subsequent research will extend this to investigate whether storage conditions affect other key properties, such as anti-oxidative stress and osteogenic effects in vitro, and in vivo biological functions, laying the foundation for the application of LELNs in the treatment of periodontitis.

## 5. Conclusions

In conclusion, we found that LELNs exhibited relative stability in both characterization and anti-inflammatory effects on periodontal ligament cells after one month of storage at different temperatures. However, particle and protein concentration of LELNs both decreased when stored at higher temperatures. Ultimately, −80 °C proved to be the optimal storage condition for preserving the structure and function of LELNs.

## Figures and Tables

**Figure 1 biomedicines-14-00099-f001:**
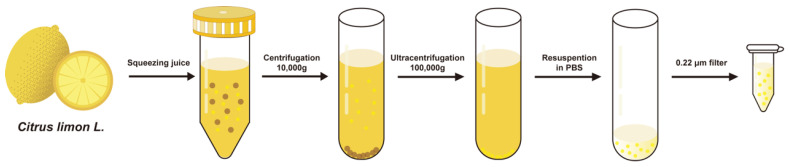
Isolation process of LELNs.

**Figure 2 biomedicines-14-00099-f002:**
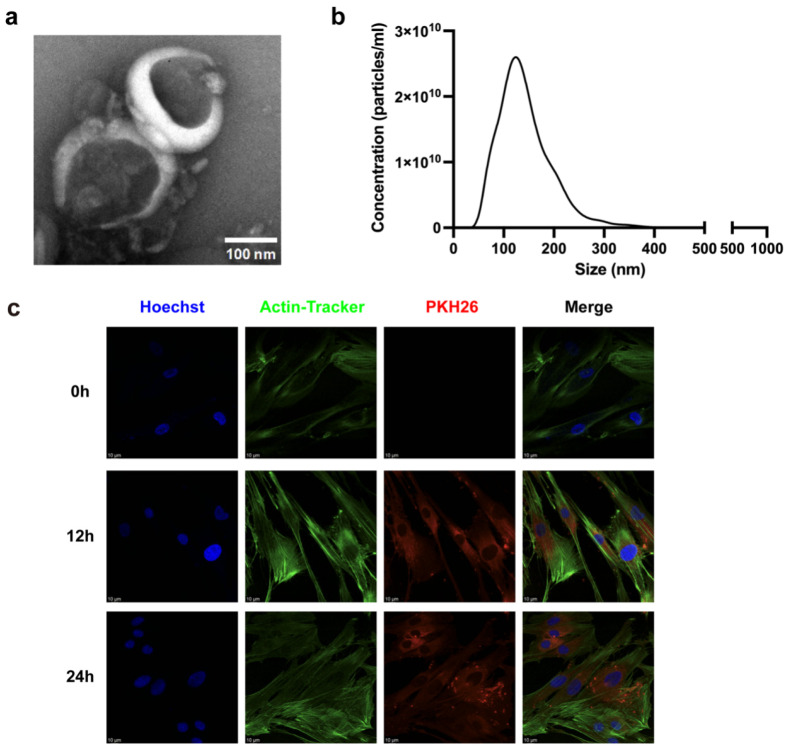
Characterization of LELNs. (**a**) TEM image showing morphology of LELNs. Scale bar = 100 nm. (**b**) NTA showing particle size distribution of LELNs. (**c**) Fluorescence microscopy images displaying the localization of LELNs in hPDLCs at 0, 12, and 24 h. LELNs were stained with PKH26 (red), the cell nucleus was stained with Hoechst (blue), and the cytoskeleton was stained with Actin-Tracker (green). Scale bar = 10 μm.

**Figure 3 biomedicines-14-00099-f003:**
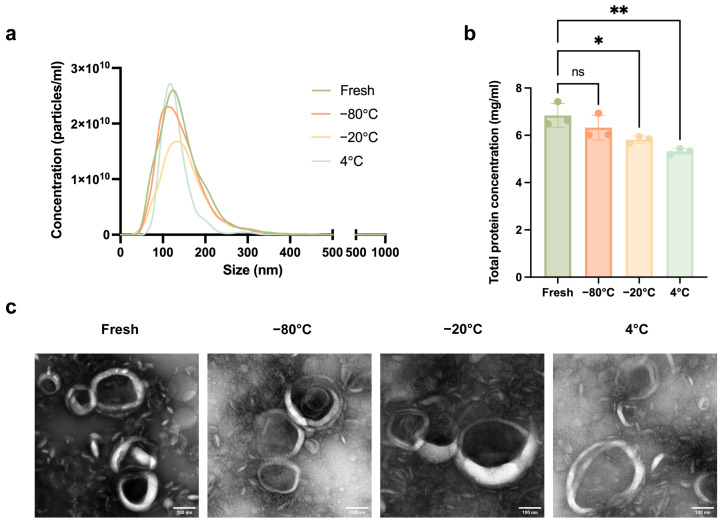
Characterization of LELNs after being stored for 1 month at different temperatures. (**a**) NTA showing particle size distribution of LELNs freshly isolated and stored at −80 °C, −20 °C, and 4 °C for one month. (**b**) BCA analysis showing protein concentration of LELNs freshly isolated and storage at −80 °C, −20 °C, and 4 °C for one month (*n* = 3). (**c**) TEM images showing morphology of LELNs freshly isolated and stored at −80 °C, −20 °C, and 4 °C for one month. Scale bar = 100 nm. All data are expressed as mean ± SD (* *p* < 0.05, ** *p* < 0.01, ns, not significant).

**Figure 4 biomedicines-14-00099-f004:**
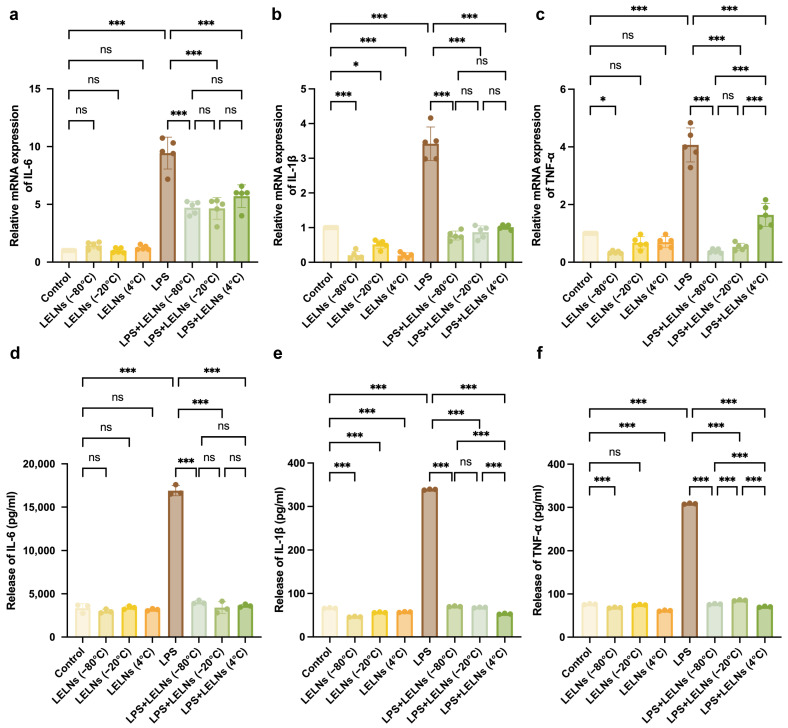
LELNs after storage for 1 month in different temperature reduce mRNA and protein levels of inflammatory factors in hPDLCs. (**a**–**c**) qRT-PCR analysis of mRNA levels of pro-inflammatory factors IL-6, IL-1β, and TNF-α in hPDLCs under different treatments (*n* = 5). (**d**–**f**) ELISA quantification of extracellular secretion of IL-6, IL-1β, and TNF-α in hPDLCs under different treatments (*n* = 3). All data are expressed as mean ± SD (* *p* < 0.05, *** *p* < 0.001, ns, not significant).

**Table 1 biomedicines-14-00099-t001:** Sequences of the primers used for qRT-PCR.

Primers	Forward (5′–3′)	Reverse (5′–3′)
GAPDH	TCATTGACCTCAACTACATG	TCGCTCCTGGAAGATGGTGAT
IL-6	GTAGCCGCCCCACACAGA	CATGTCTCCTTTCTCAGGGCTG
IL-1β	GCCAGTGAAATGATGGCTTATT	AGGAGCACTTCATCTGTTTAGG
TNF-α	CTCATCTACTCCCAGGTCCTCTTC	CGATGCGGCTGATGGTGTG

**Table 2 biomedicines-14-00099-t002:** NTA showing particle concentration and average particle size of LELNs freshly isolated and stored at different temperatures for one month.

	Fresh	−80 °C	−20 °C	4 °C
Concentration (particles/mL)	2.8 × 10^12^	2.6 × 10^12^	2.0 × 10^12^	1.8 × 10^12^
Average particle size (nm)	142.5 ± 2.4	142.8 ± 1.8	151.7 ± 3.9	131.0 ± 2.3

## Data Availability

The raw data supporting the conclusions of this article will be made available by the authors on request.

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
