# Peer review of "Characterization and Anti-Inflammatory Effects on Periodontal Ligament Cells of Citrus limon-Derived Exosome-like Nanovesicles Under Different Storage Temperatures"

_biomedicines, 2026, doi:10.3390/biomedicines14010099_

Round 1

Reviewer 1 Report

Comments and Suggestions for Authors

Following are the comments for the author while revising the manuscript.

  • The introduction provides background but does not clearly state a hypothesis, such as stability differences and what biological mechanisms.
  • For clinical or industrial application, storage assessments should last longer than the 3–12 months tested; this significantly limits the real-world relevance of the study's findings.
  • The characterization lacks plant-derived EV markers. TEM + NTA are not enough for confirmation. Authors perform Western blot/flow cytometry for surface markers.
  • Although the authors claim particle size is “similar,” the values vary considerably -Fresh-142.5 nm, −20°C- 151.7 nm, 4°C-131 nm. This 20 nm difference is significant in EV characterization.
  • Also, multiple technical replicates are not shown and polydispersity index (PDI) is not provided
  • The authors state samples underwent “only one freeze-thaw cycle,” but this is not experimentally shown Fresh sample vs. one freeze-thaw or Fresh sample vs. 2–3 freeze-thaw cycles
  • Only one concentration of LELNs was used and to validate anti-inflammatory action, the authors should test at least 3 concentrations based on the IC50 values
  • 1 μg/mL LPS for 24 h may induce cytotoxicity rather than simple inflammation.
  • n = 3 is very low for qRT-PCR, it is technical vs biological replicates?  Error bars appear large and inconsistent. Multiple comparisons correction is not clearly described.

Reviewer 2 Report

Comments and Suggestions for Authors

The manuscript titled Characterization and Anti-inflammatory Effects on Periodontal Ligament Cells of Citrus limon-Derived Exosome-Like Nanovesicles under Different Storage Temperature presents a relevant investigation into the stability of plant-derived nanovesicles for potential therapeutic applications in periodontitis. While the study provides valuable data regarding the preservation of biological activity at various temperatures, there are several areas where the manuscript could be improved to enhance its scientific rigor and clarity.

One primary concern relates to the presentation of the results from the Nanoparticle Tracking Analysis and Transmission Electron Microscopy because the authors report similar average particle sizes across groups while their TEM images and text descriptions highlight significant aggregation and heterogeneity in samples stored at higher temperatures. The authors should provide a more detailed reconciliation of why the NTA data might not fully capture the structural changes or aggregation clearly visible in the microscopic observations. Furthermore, the Discussion section would benefit from a more profound exploration of the underlying biochemical mechanisms that allow LELNs to maintain their anti-inflammatory properties despite the observed morphological degradation and protein loss at four degrees Celsius. Regarding the experimental design, the study is currently limited by a relatively short storage duration of only one month and does not account for other critical stability factors such as the impact of multiple freeze-thaw cycles or the use of cryoprotectants. It is recommended that the authors acknowledge these limitations more explicitly or ideally include preliminary data on longer-term stability to strengthen their conclusions regarding the optimal storage conditions. Additionally, the manuscript contains some minor technical errors such as a placeholder for a figure reference on page five and repetitive funding information in the acknowledgments section that must be corrected during the revision process. Expanding the scope of biological assays to include antioxidant or osteogenic effects would also provide a more comprehensive understanding of how storage affects the multifaceted therapeutic potential of LELNs in the context of periodontal treatment.  

Round 2

Reviewer 1 Report

Comments and Suggestions for Authors

The authors made the changes suggested by the reviewer.